# Yoyo Dieting, Post-Obesity Weight Loss, and Their Relationship with Gut Health

**DOI:** 10.3390/nu16183170

**Published:** 2024-09-19

**Authors:** Kate Phuong-Nguyen, Sean L. McGee, Kathryn Aston-Mourney, Bryony A. Mcneill, Malik Q. Mahmood, Leni R. Rivera

**Affiliations:** 1Institute for Mental and Physical Health and Clinical Translation (IMPACT), School of Medicine, Deakin University, Waurn Ponds, VIC 3216, Australia; sean.mcgee@deakin.edu.au (S.L.M.); k.astonmourney@deakin.edu.au (K.A.-M.); bryony.mcneill@deakin.edu.au (B.A.M.); 2School of Medicine, Deakin University, Waurn Ponds, VIC 3216, Australia; malik.mahmood@deakin.edu.au

**Keywords:** yoyo dieting, weight loss, obesity, gut microbiome

## Abstract

Excessive body weight is associated with many chronic metabolic diseases and weight loss, so far, remains the gold standard treatment. However, despite tremendous efforts exploring optimal treatments for obesity, many individuals find losing weight and maintaining a healthy body weight difficult. Weight loss is often not sustainable resulting in weight regain and subsequent efforts to lose weight. This cyclic pattern of weight loss and regain is termed “yoyo dieting” and predisposes individuals to obesity and metabolic comorbidities. How yoyo dieting might worsen obesity complications during the weight recurrence phase remains unclear. In particular, there is limited data on the role of the gut microbiome in yoyo dieting. Gut health distress, especially gut inflammation and microbiome perturbation, is strongly associated with metabolic dysfunction and disturbance of energy homeostasis in obesity. In this review, we summarise current evidence of the crosstalk between the gastrointestinal system and energy balance, and the effects of yoyo dieting on gut inflammation and gut microbiota reshaping. Finally, we focus on the potential effects of post-dieting weight loss in improving gut health and identify current knowledge gaps within the field, including gut-derived peptide hormones and their potential suitability as targets to combat weight regain, and how yoyo dieting and associated changes in the microbiome affect the gut barrier and the enteric nervous system, which largely remain to be determined.

## 1. Introduction

For decades, we have witnessed the increasingly deleterious effects of obesity, which constitutes a major health and economic crisis in the modern world [1]. Obesity and overweight have become a growing problem affecting 52% of the world’s adult population [2,3,4,5], a figure that is expected to continue to rise [1,6]. Obesity presents as an emerging global epidemic leading to more than 2.8 million deaths annually [7] from diseases such as cardiovascular diseases [8,9], type 2 diabetes [10], fatty liver disease [11,12] and various systemic metabolic syndromes [13,14,15,16,17]. Despite much research and clinical effort to identify optimal solutions to control body weight, sustained weight loss is virtually impossible for most individuals with obesity, as most or all of the lost weight often returns within 5 years [18,19,20,21,22,23,24,25,26,27,28,29,30,31,32,33,34,35]. While weight loss remains a key therapeutic approach for many metabolic diseases [36,37], many individuals experience multiple cycles of recurrent obesity increasing their risk of developing significant obesity-related comorbidities [28,38].

### 1.1. Yoyo Dieting: Cyclic Weight Regain after Weight Loss

The weight loss journey of every individual is unique and not everyone is successful in losing weight. However, a common pattern for most individuals attempting to lose weight is an initial period of rapid weight loss, which gradually slows, before subsequent weight regain. This occurs despite every effort to adhere to weight management strategies [39]. This phenomenon is called weight cycling or “yoyo dieting” [40]. For many individuals who are medically advised to lose weight, yoyo dieting makes the weight loss journey even more challenging as the initial weight loss often returns and the weight loss process must be restarted. After multiple failed attempts to achieve sustained weight loss, many people refuse to continue their weight loss journey believing that weight loss is an impossible mission [41].

The likelihood of weight regain is greatest in the period following initial weight loss [42,43]. Yoyo dieting does not only affect people with excessive weight, as studies suggest that frequent on–off dieting also triggers weight regain in people who are not overweight or obese [44,45,46]. Dieting has become a popular norm for many people, including individuals with a healthy body weight (body mass index [BMI] between 18.5–24.9 kg/m^2^), and those underweight (BMI < 18.5 kg/m^2^) [47,48], who feel the urge to lose weight to achieve their desired body image [49]. However, as is usual with yoyo dieting, these individuals end up gaining all or even more weight after each cycle of weight loss [50,51,52]. Therefore, while the prevalence of obesity has been rising, yoyo dieting requires consideration because it places individuals at greater risk of the progression to obesity [53], and the development of associated chronic complications [54,55,56,57,58]. 

Taken together, although the link between yoyo dieting and eventual weight gain is well established, a complete picture of the mechanisms driving this response and the strategies to prevent it are still being explored. Weight gain following yoyo dieting is complex and involves changes in peripheral hormones regulating energy balance and metabolic adaptation to weight loss. The gut also plays an important role in body weight regulation and could be a key factor in weight regain after yoyo dieting.

### 1.2. The Relationship between Obesity, the Gut, and the Susceptibility to Weight Regain Is Potentially Due to Changes in the Gut Microbiota

There is established evidence from both human [59,60] and animal studies [61,62] implicating the gut in obesity development, largely due to an increase in the potentially harmful microbiota (termed ‘gut dysbiosis’ [63]). Gut dysbiosis (e.g., enriched *Fusobacterium* and reduced *Oscillospira*, Ruminococcaceae, Coriobacteriaceae, *Odoribacter splanchnicus*, *Akkermansia muciniphila* and *Bifidobacterium longum*) is strongly associated with negative metabolic outcomes in individuals with overweight and obesity [60,64] compared to controls (e.g., enriched *Akkermnansia muciniphila*, *Alistipes indistinctus*, *Odoribacter splanchnicus*, *Clostridium* sp. *CAG:413*, *Intestinimonas butyriciproducens*, and *Bifidobacterium longum*) [64]. Studies using germ-free mice provide evidence for a direct role of the microbiota in the metabolic alterations typically seen in individuals with obesity. For example, germ-free mice colonised with microbiota from obese mice (enriched *Bacteroides uniformis*, *Parabacteroides merdae*, *Alistipes putredinis*, *Eubacterium rectale* and reduced *Bacteroides thetaiotaomicron*) extracted more calories from their food and had reduced energy expenditure, leading to increased adiposity when compared with germ-free mice colonised with microbiota from lean mice [59,65]. These data indicate that the microbiota influences the efficiency of both nutrient absorption and calorie expenditure [17], strongly implicating a causative role of the gut microbiota in obesity pathogenesis [66].

The relationship between yoyo dieting, post-obesity weight loss, and gut health has been largely overlooked. To date, only a small number of studies have explored the effect of weight regain after weight loss on the gut microbiome [61,62]. These studies have indicated that yoyo dieting results not only in enhanced weight gain after dieting but also in potential long-term alterations in gut microbiota composition [61]. Given that most people who lose weight end up regaining all or even more weight than was previously lost, it is evident that this is an area that warrants further investigation. This review will focus on the influence of the gut on body weight changes during yoyo dieting and the relationship between fluctuating body weight, gastrointestinal inflammation, and gut microbiome alterations. Moreover, this review also explores the potential benefits of weight loss in influencing positive changes in the gut and identifies current knowledge gaps that need to be investigated.

## 2. The Influence of the Gut in Weight Regain after Weight Loss

While there is not yet a complete picture of all the mechanisms behind weight recurrence after loss, evidence suggests it is due to alterations in peripheral peptides that regulate energy balance and metabolic adaptation to weight loss. These changes might contribute to reduced energy expenditure and an increased drive to eat to counteract the energy gap created during dieting [42], hence placing individuals at a higher risk of obesity. 

### 2.1. Gut Peptide Hormones That Regulate Energy Balance

Regulation of energy balance involves adjustment of energy intake, energy expenditure and energy storage [67] and is the main driver of body weight change. Energy balance is also the key concept describing the pathogenesis and prevalence of obesity and its comorbidities and is an important consideration in the development of possible treatments for weight control [48]. Energy balance is regulated not only by the central nervous system but is also strongly influenced by various peripheral signals arising from the gastrointestinal tract, pancreas, and adipose tissue, collectively acting to either stimulate or restrict energy intake [68,69]. These peripheral signals, many of which are peptide hormones, regulate energy balance by directly or indirectly controlling the activity of orexin and anorexic neurons within integrating brain regions such as the hypothalamus [70]. 

Peptide hormones secreted by the gut act via several pathways to modify energy status. These gut peptide hormones can act locally on gut peptide receptors expressed on vagal or spinal afferent nerve terminals innervating the gut to activate gut-brain neuronal signalling [71]. Moreover, they also act indirectly via receptors on intrinsic neurons of the enteric nervous system to relay neuronal signalling to afferent nerves [72]. The gut-derived hormones peptide tyrosine-tyrosine (PYY) and glucagon-like peptide 1 (GLP-1) delay gastric emptying [73], cholecystokinin (CCK) promotes bile production, pancreatic and gastric secretions [74], and oxyntomodulin (OXM) increases acid secretion in response to food intake [75]. These peptide hormones are also known to regulate feeding behaviour [76], specifically promoting appetite suppression and meal termination [77]. In contrast, the hunger gut-derived peptide hormone, ghrelin, increases gastric emptying, promotes appetite stimulation and lipogenesis, reduces lipid oxidation, and accelerates the use of carbohydrates as a source of energy while sparing fat [78]. Collectively, gut-derived peptide hormones are essential in the direct regulation of hunger/satiety states and energy consumption in humans. Increased appetite that drives energy intake to surpass energy expenditure will increase fat accumulation and result in body weight gain [79]. This is thought to be partly but critically associated with a gut peptide hormone profile that increases energy intake and reduces energy expenditure [80] although it is not yet clear if they are the trigger or consequence of obesity [81]. 

In addition to these signals from the gut, peptide hormones secreted by adipose tissue act via neuroendocrine mechanisms to convey information on energy status from the periphery to the brain. One such peptide hormone, leptin, is responsible for relaying information regarding hunger and satiety to the hypothalamus [82]. Circulating leptin levels increase in proportion to adipose tissue mass, which can result in a state of leptin resistance in obesity [83,84,85,86]. Adiponectin is a hormone synthesised and secreted exclusively by adipose tissue that suppresses appetite [87] and affects thermogenesis, with its receptors expressed in multiple peripheral tissues (such as liver and muscle) and centrally, in the hypothalamus [88]. In contrast to leptin, adiponectin circulating concentrations are reduced in obesity [89]. Furthermore, the pancreas also plays a crucial role in secreting hormones that affect energy homeostasis and appetite control, namely insulin, amylin, GLP-1, pancreatic polypeptide (PP), and glucose-dependent insulinotropic polypeptide (GIP). These feeding-relevant neuroendocrine signals bind to receptors in the hypothalamus and hindbrain to simultaneously regulate energy homeostasis, normalise the internal milieu during nutrient influx [90], regulate the metabolism of macronutrients [91], suppress appetite [92], and improve energy storage due to adipogenesis [93]. 

The regulation of appetite, body weight, and fat mass involves a complex coordination of various peripheral peptides and an imbalance of these peptide hormones might contribute to weight regain after loss. Results from several studies indicate that peripheral responses to calorie deficit promote increased hunger and a lower metabolic rate thereby enhancing susceptibility to weight relapse [77,94]. This involves decreases in gut-derived-satiety peptide hormones GLP-1 [77,95], PYY [77,96], and CCK [77,94,97], adipose-derived leptin, and the pancreas peptide hormones insulin and amylin [77,94]; in contrast with increased levels of the hunger peptide hormones ghrelin and PP [77,94]. There is evidence that these changes persist as individuals attempt to maintain reduced body weight, even after the cessation of active weight loss [94]. The continued low energy intake and minimal fat mass indicate energy unavailability, leading to a homeostatic endocrine response aimed at conserving energy and promoting energy intake [42,61,62,98,99,100]. These represent a critical area of ongoing research for obesity pathogenesis and a major challenge to developing strategies for the treatment of this relapsing disorder [68]. 

While specific mechanisms underlying the reduction in gut-derived satiety hormones following calorie deficit-derived weight loss are not fully understood, increasing evidence suggests it is possibly associated with various alterations related to the gut. Of note, fat is an essential macronutrient stimulant in the small intestine for PYY and CCK release [101]. However, most calorie-deficit diets reduce fat intake, therefore potentially reducing circulating PYY and CCK. Additionally, the gut epithelial cells that produce the incretin peptides, enteroendocrine cells (EECs) [102] located throughout the gastrointestinal tract, undergo constant cellular renewal, being replenished with new EECs formed every 5–7 days [103]. One of the important factors impacting intestinal epithelial cell differentiation and especially EEC renewal is the expression of Inhibitor of differentiation/DNA-binding protein 2 (Id2) transcription factor expressed mainly in the EECs themselves [104] and adipose tissue [105]. Dramatic loss of fat mass is often associated with a reduced expression of Id2 [106]; this potentially leads to decreased EEC renewal, hence reducing satiety hormones commonly seen shortly after weight loss. The reduction of Id2 in the EEC is associated with significantly reduced levels of the satiety peptide hormone PYY and the elevated hunger peptide hormone Ghrelin [104]. A study by Wölnerhanssen et al. [107] indicated that the total number of EECs expressing satiety hormones was significantly reduced during obesity but returned to control levels 3 months post-obesity. This suggests that the number of EECs expressing satiety hormones could be reduced in obesity but eventually restored in the long term if weight loss is substantial. Moreover, it is also worth noting that these findings were taken from a small sample size (*n* = 14) of individuals with morbid obesity [107]. These indicate that further longitudinal research with a larger study population is warranted to further verify these data. Another factor that is associated with reduced satiety hormone production after a calorie-restrictive diet is the reduced relative abundance of short-chain fatty acids (SCFAs)-producing microbiome. SCFAs are metabolites produced by specific colonic microbes [108] and are potent stimuli for the secretion of PYY [109], GLP-1 [110], and CCK [111]. After weight loss, the relative abundance of several SCFA-producing microbiota was found to be reduced, such as *Roseburia intestinalis* and *Eubacterium rectale* [112,113], *Faecalibacterium prausnitzii* [113], and *Agathobacter rectalis* [114]. This might help to explain how the gut microbiome influences the imbalance in satiety hormone production that contributes to the yoyo effect, but more studies are required to fully elucidate the mechanisms involved.

The potential benefits of targeting peripheral hormones as an approach for weight management are best highlighted by bariatric surgery. Bariatric surgery alters gastrointestinal anatomy impacting food intake and/or nutrient absorption [115], and is so far the most effective treatment for substantial weight loss. One common finding after bariatric surgery is a sharp rise of many satiety peptide hormones after meals, such as CCK and GLP-1, leading to increased satiety and decreased food reward [116,117]. Evidence suggests that this is due to the accelerated delivery of nutrients down the gastrointestinal tract, where the majority of EECs are situated [116]. Persistent exposure to rapid nutrient entry can lead to enhanced nutrient sensing within the gut and an increased number of satiety hormone-containing ECCs [118] leading to an increased secretion of CCK [116], GLP-1 [118]. Collectively, gut hormones have been demonstrated as key players regulating weight loss in anti-obesity surgeries [119,120,121]. Post-obesity weight loss increases both hunger and food reward, in part due to an imbalance in satiety-signalling hormones. This hormonal imbalance likely contributes to the yoyo effect, as a result of the body’s attempt to re-establish a body weight set point after a period of negative energy balance [122]. 

New weight loss therapeutics, including GLP-1 receptor agonists (slowing down gut transit [123]—Liraglutide [124] and Semaglutide [125]), dual GIP and GLP-1 receptor agonist (Tirzepatide [126]), and pancreatic and gastric lipase inhibitors (inhibiting dietary triglycerides digestion and absorption in the intestine [127,128]—Orlistat [129]) either signal through peripheral hormone mechanisms or have effects on peripheral hormones that regulate energy balance. To date, these medications are approved for long-term usage [130,131]. However, they are often only used on a short-term basis [131] mainly due to high costs [132], adverse side effects [131,132,133] that most commonly involve gastrointestinal symptoms [132,134], or insufficient response [131,132,133]. This highlights that gut hormones remain a critical focus in weight management [119,120,121].

### 2.2. Sympathetic Nervous System Regulates Energy Expenditure and Thermogenesis as a Metabolic Adaptation to Weight Loss

A decrease in energy intake causes a decrease in energy expenditure and vice versa. This is mainly influenced by non-adaptive modifications in the metabolically active component of the body known as fat-free mass. However, weight change does not always precisely follow predictions based on the calculation of energy imbalance. This is explained by alterations in the sympathetic nervous system (SNS) activity resulting in adaptive metabolism that is associated with weight gain in obesity [135]. Further, reduced activation of β-adrenergic signalling (stimulating lipolysis and thermogenesis [136]) leads to the promotion of weight gain after loss [137]. In particular, the SNS plays a key role in regulating metabolic homeostasis and is central in monitoring daily energy expenditure via the control of resting metabolic rate and induction of thermogenesis in response to physiologically related stimuli (hyperinsulinemia, energy states, food intake, dietary nutrients and cold exposure). Activation of the SNS in the liver, pancreas, and adipose tissue can also provoke acute catabolic responses (lipolysis and glycogenolysis) [138] that play an essential role in determining adaptive thermogenesis. Thermogenesis (metabolic heat production, mainly in brown adipose tissue [139]) plays a key role in energy expenditure [140,141]. Adaptive thermogenesis restricts modifications in energy stores in response to changing energy balance. It creates an ideal physiological environment to store energy and encourage weight regain by reducing energy expenditure beyond what can be predicted by the loss of fat mass and fat-free mass, hence preventing changes in body composition [142,143]. Changes in adaptive thermogenesis, which are known to be associated with metabolic efficiency [144] and weight change [145], could play an important role in metabolic adaptation to weight loss leading to the yoyo effect.

Taken together, weight loss is associated with changes in the regulation of the SNS, adaptive thermogenesis, and adaptive alterations in satiety gut hormones. These may partially explain insufficient weight loss outcomes and the high risk of weight regain in individuals with obesity and yoyo dieting [142,143].

## 3. Yoyo Dieting and Gut Inflammation

### 3.1. Links between Yoyo Dieting and Gut Inflammation

It is well established that chronic low-grade systemic inflammation is a hallmark of obesity [146,147,148,149,150]. For example, circulating adipose-derived cytokines, such as tumour necrosis factor-α (TNF-α) or interleukin-6 (IL-6) are elevated in obesity [151,152,153,154,155]. Yoyo dieting also promotes inflammatory responses [156,157] upon weight regain due to increased M1 proinflammatory macrophage activation and monocyte chemotactic protein 1 (MCP-1) proinflammatory cytokine production [156]. This suggests a potential contribution to an obesogenic memory phenotype [156]. Elevated levels of a number of cytokines have been reported after the 2nd weight gain of yoyo dieting, including IL-6 and TNF-α in the cell culture media primed with bone marrow-derived macrophage of yoyo dieting mice [158], interferon-gamma (IFN-γ) in epididymal adipose tissue, and signal transducer and activator of transcription 3 (STAT3) and nuclear factor kappa-light-chain-enhancer of activated B cells (NF-kB) in liver [156]. With respect to inflammation specifically within the gut, upregulated intestinal immune control leading to increased gut inflammation and permeability are known characteristics of obesity [150,159,160,161,162,163,164,165,166,167,168,169,170]. Although there are limited studies specifically looking at the direct effect of yoyo dieting on gut inflammation, findings from several animal studies provide insights into potential links between yoyo dieting and gut inflammation. The effects of yoyo dieting have been tested experimentally by frequently switching from high-fat to low-fat diets, and examining the associated effects on gastrointestinal structure, function, and inflammation. For example, a study by Li et al. [171] demonstrated that mice fed a yoyo diet showed significantly lower C1q/tumour necrosis factor-related protein-3 (CTRP3) expression in epididymal adipose tissue in comparison to mice fed an HFD only. Reduced expression of CTRP3 is associated with a greater risk of intestinal inflammation [172,173]. Recently, another study observed a significant increase in CD11c (M1 proinflammatory macrophage marker) and MCP-1 in the epidydimal adipose tissue of mice fed a yoyo diet (HFD, Control, and then HFD) compared to those fed only one cycle of HFD (Control, and then HFD) [156]. MCP-1, one of the key chemokines of inflammation, is predominantly present in the gastric mucosal epithelium [174,175] and is known to be significantly enhanced in inflamed intestinal biopsies of IBD individuals and Caco-2 colon carcinoma cells [174,176,177]. Of note, increased M1 macrophages and MCP-1 expressions are strongly associated with elevated chronic intestinal inflammation in IBD [174,178,179,180,181]. Collectively, these data suggest that weight regain after loss in yoyo dieting contributes to increased intestinal inflammation in animals. However, there is a lack of human studies on yoyo dieting and gut inflammation, highlighting the need for further human studies in order to better translate these findings.

### 3.2. Post-Obesity Weight Loss as a Potential Approach to Reduce Gut Inflammation in Humans

There are currently limited studies looking at the direct effects of yoyo dieting on gut inflammation. However, the current literature suggests that diet-induced weight loss is a potential approach to reduce gut inflammation in humans. Weight loss after obesity results in significant anti-inflammatory effects in the gut. One of the early studies was conducted by Pendyala et al. [182] who assessed the levels of proinflammatory cytokines and gene expression profiles in rectosigmoid mucosal biopsies from a small human cohort of female individuals with obesity who consumed a low-calorie diet to achieve ≥8% loss of baseline weight [182]. Weight loss was correlated with significant declines in rectosigmoid mucosal TNF-α, IL-1β, IL-8 and MCP-1, suggesting reduced colonic inflammation. Moreover, transcriptomics of the colon reported that the calorie-restricted diet and weight loss significantly downregulated expression of the proinflammatory IL-8, implying a protective effect on the mucosal barrier [177,183,184]. Additionally, genes encoding for circulating peripheral peptides associated with weight loss were downregulated, including PYY and vasoactive intestinal peptide (VIP). The reduction in VIP levels is likely to be beneficial because elevated VIP is considered to be a biomarker for an aggravated state of colonic inflammatory conditions, as seen in inflammatory bowel disease (IBD) [185,186]. However, reduced PYY is associated with faster gastric emptying [187] and reduced satiety, which promotes food intake [188] and reduces energy expenditure [189] together driving positive energy balance. This is characteristic of energy balance hormone dysregulation that attempts to restore body weight upon weight loss [42]. Additionally, a human study by Ott et al. [114] investigated the effects of a 28-day caloric restriction for weight loss in female individuals with obesity and demonstrated decreased high-sensitivity CRP (inflammation biomarker) and paracellular gut permeability markers (polyethylene glycol 1500 and zonulin) in urine, and reduced chemerin (inflammation marker) in plasma. These results signify that weight loss after obesity reduces intestinal inflammation and improves barrier function [190,191]. Collectively, the current literature suggests that weight loss after obesity improves gut health. However, these findings were taken from a small sample size (*n* = 20) of women participants, indicating that further research is warranted to explore potential sex differences in responses to yoyo dieting. 

Taken together, current findings suggest an association between post-obesity weight loss and anti-inflammatory effects on the gut. Animal studies suggest the health benefit of reduced inflammation in the gut during the weight loss phase (of yoyo dieting), but this might not persist once the weight rebounds [156,171]. Moreover, while the majority of human studies also indicate the potential benefits of post-obesity weight loss in improving gut inflammation, no clinical investigation has been conducted to examine changes in gut inflammation once the lost weight is regained. Therefore, more studies, especially in humans, are needed to have a better understanding of the potentially proinflammatory effects of yoyo dieting on gut health. 

## 4. Yoyo Dieting and Gut Microbiome

The gastrointestinal tract is host to an extraordinarily abundant and diverse community of microorganisms, including bacteria, archaea, viruses, fungi, and protozoa, collectively known as the gut microbiota. It is widely recognised that the gut microbiota exerts a vital role in influencing the host health [192,193] and there is a strong association between gut microbiota dysbiosis and obesity, which has been reported in animal and human studies [59,61,62,65,194]. There is also emerging evidence suggesting an important role for the gut microbiome in influencing weight recurrence after dieting. While there have been conflicting findings, most studies indicate that yoyo dieting reduces alpha-diversity, and alters microbiota composition, such as the Firmicutes:Bacteroidetes ratio. Alterations in gut microbiota diversity, abundance, and composition are closely linked to changes in metabolism, as well as the development of obese/lean phenotypes. Several studies have investigated yoyo dieting and potential interventions, aiming not only for weight loss but also for the restoration of normal gut microbiota. These investigations have sought to understand how to recalibrate the microbiota of individuals struggling with obesity as a method to alleviate associated metabolic complications. 

### 4.1. Different Gut Microbiota Profiles in Yoyo Dieting

Several animal studies have indicated an association between yoyo dieting and lower alpha diversity (distribution of microbiota richness and evenness [195]) and alterations in gut microbiota composition and relative abundance [61,62,196]. One of the earliest animal studies investigating the influences of yoyo dieting on the microbiome was conducted by Thaiss et al. [61]. In their study, mice in the yoyo group were exposed to two HFD and LFD cycles with each diet phase continued until these mice were a similar weight to the obese (HFD) and lean (LFD) control mice. In either the 1st or 2nd yoyo cycle the microbiota composition of the mice during obesity (HFD feeding period) significantly differed from that of the LFD control mice. However, even after achieving successful weight loss with similar body weights to lean control mice, the microbiota composition of yoyo mice remained in an intermediate state that was distinct from control mice. In fact, it took 21 weeks after successful weight loss (5 times longer than the last dieting period) for the microbiota composition of the yoyo dieting mice to return to the control state. Therefore, although short-term weight loss after obesity is associated with immediate metabolic health improvements, the gut microbiota requires longer-term maintenance of weight loss to return to a non-obese state [61]. This suggests that gut microbiota could increase the susceptibility to weight regain immediately after weight loss. The specific changes in the microbiome during obesity and after initial weight loss included consistently lower alpha diversity and relative abundance of *Christensenella* spp. and *Lactobacillus reuteri*. *Christensenella* spp. is associated with leanness in both animal and human studies [197,198,199,200,201] and is also associated with a reduced risk of IBD [202,203]. Hence, the reduction of *Christensenella* spp. could have direct negative effects on body weight and inflammation [201,204]. Additionally, *Lactobacillus reuteri* is a well-known commensal gut symbiont commonly used as a probiotic [205,206] due to its ability to inhibit pathogenic microbiota colonisation [207], ameliorate inflammation [208,209,210], and improve intestinal barrier function [211]. Therefore, the dramatic loss in the relative abundance of *Lactobacillus reuteri* could have deleterious effects on these functions. 

Another study by Kawashima et al. [196] showed that mice fed a yoyo diet had significantly lower alpha diversity and a higher relative abundance of Firmicutes (genus *Ruminococcus*) and Proteobacteria (genus *Desulfovibrio*) compared to mice fed a control diet. Enrichment of *Ruminococcus* is likely unfavourable because this anaerobic Gram-positive bacteria genus [212] has been associated with an increased risk of Crohn’s disease [213,214] and metabolic syndrome [215]. Moreover, the increased relative abundance of *Desulfovibrio* is also not likely to be beneficial because this anaerobic Gram-negative rod/sulphate-reducing bacteria [216] has been associated with ulcerative colitis [217,218] and Parkinson’s disease [219]. 

In another recent mouse study by Humblot et al. [62], the long-term effects of yoyo diets on gut microbiota composition were investigated. Mice fed a yoyo diet ending with LFD had higher alpha diversity than those ending with HFD. There were no significant differences in alpha and beta diversity between obese mice fed HFD only and yoyo mice ending with HFD, and between control mice fed LFD only and yoyo mice ending on LFD. Overall, their results indicated that obese mice and yoyo mice ending with HFD had a higher proportion of Firmicutes (classes Erysipelotrichia and Bacilli) and upregulated functional microbiota genome dedicated to amino acid synthesis, while control mice and yoyo mice ending with LFD had a higher proportion of phylum Bacteroidetes and genus *Lactobacillus*, and a higher proportion of genes coding for carbohydrate metabolism and biosynthesis of B group vitamins. An increased relative abundance of Erysipelotrichia and Bacilli is associated with increased intestinal inflammation and is considered a biomarker of IBD [220], suggesting a putative role in the obesity phenotype. Conversely, a greater relative abundance of *Lactobacillus* is strongly associated with improved intestinal barrier function [221], reduced intestinal lipid absorption [222,223], and a representative probiotic for IBD [224] and liver diseases [225]. Their results suggest that the microbiota function and composition of mice fed yoyo diets do not have impacts beyond the diet period and that yoyo dieting would not have any long-term consequences. 

A summary of yoyo dieting studies affecting the gut microbiota using animal models is listed in Table 1.

Taken together, current evidence from animal studies suggests that yoyo dieting results in potential impacts on the gut microbiome. However, conflicting results on the residual changes of the gut microbiome during and after yoyo dieting warrants further research, in particular determining whether long-term gut dysbiosis is a result of yoyo dieting and possibly a key factor for weight regain susceptibility.

### 4.2. Will the Gut Microbiome Be the Next Target to Prevent Weight Regain?

There are limited studies that have investigated yoyo dieting in humans. However, some clues can be taken from animal studies exploring the effect of dietary interventions targeting weight loss and clinical studies exploring the effects of post-obesity weight loss, which indicate significant changes in gut microbiota composition, leading to potentially positive signs of improving gut and metabolic health. 

#### 4.2.1. Dietary Interventions Enhance Weight Loss and Alter Gut Microbiota in Animal Studies

Several animal studies investigating dietary interventions for weight loss during yoyo dieting suggest notable alterations in gut microbiota composition, hinting at potential improvements in gut and metabolic health. There has been growing evidence in both in vivo [226] and in vitro studies [227,228,229] suggesting that flavonoids, a type of bioactive compound derived from beverages, vegetables and fruits [230], could improve weight management. For example, a study by Thaiss et al. [61] showed that flavonoid treatment resulted in less weight regain compared to vehicle-treated mice during yoyo dieting. Moreover, the administration of flavonoids was associated with a dramatically elevated expression of the major thermogenic factor uncoupling protein 1 (UCP1) in brown adipose tissue. Apart from the weight control effect, the increase in intestinal flavonoids is likely to be beneficial for gut health because these dietary polyphenols are mostly processed by the gut microbiota to exert effective roles with anti-oxidant and anti-inflammatory factors [231,232]. Furthermore, increased intestinal flavonoid is also associated with amelioration of gut microbiota dysbiosis [233,234,235,236], reduced risk of colorectal cancer [237,238,239,240,241,242] and inhibition of *Helicobacter pylori* infection-associated gastric cancer [243]. 

Another study by Kawashima et al. [196] explored the effect of yoyo dieting and the use of daisaikoto (a Japanese traditional herb-based medicine used as a treatment for menstrual pain [244] and fatty liver disease [245,246,247]) in preventing weight regain and reversing gut dysbiosis. In this study, mice were fed a cyclic HFD interspaced with LFD, with and without supplementation of 3% daisaikoto during the LFD feeding period only. Mice fed a yoyo diet supplemented with daisaikoto had significantly lower body weight, visceral fat, and weight regain upon a subsequent HFD feeding period in comparison with mice fed a yoyo diet only. Additionally, yoyo mice fed daisaikoto had a significantly higher relative abundance of Bacteroidetes and lower Firmicutes than those without daisaikoto. Although there are conflicting findings, a Firmicutes:Bacteroidetes ratio greater than 1 is often associated with the obese phenotype, while a ratio lower than 1 is associated with a lean phenotype [248,249,250,251]. This might imply that daisaikoto supplementation in yoyo dieting mice is beneficial for weight loss potentially due to positive changes in the microbiome. Yoyo mice fed daisaikoto had similar relative abundances of *Ruminococcus* and *Desulfovibrio* compared to control mice and lower than yoyo mice fed no daisaikoto. Reduced relative abundances of *Ruminococcus* and *Desulfovibrio* are likely to be beneficial because these are associated with a reduced risk of IBD development [217,252,253].

Additionally, a recent animal study by Zhong et al. [254] suggested that, following weight loss, a high protein diet during weight regain reduced lipid absorption in the intestine and reduced fat accumulation compared to a standard protein diet during weight regain. Mice fed a high protein diet had higher alpha diversity and a lower relative abundance of *Lactobacillus murinus* Lam-1 than those fed with a normal protein diet. Remarkably, their finding confirmed that enrichment of *Lactobacillus murinus* Lam-1 is associated with enhanced intestinal lipid absorption and fat accumulation. These data imply that weight loss potentially endorses the enrichment of intestinal *Lactobacillus murinus* Lam-1 leading to dramatic weight regain and that the weight recurrent issue might be tackled by a diet high in protein. 

#### 4.2.2. Post-Obesity Weight Loss May Be Beneficial in Improving Gut Microbiota in Humans

Clinical studies have shown that post-obesity weight loss is associated with significant changes in gut microbiota composition, leading to potentially positive signs of improving gut and metabolic health. For example, a human study by Ott et al. [114] investigated the effects of a one-month caloric restriction in female individuals with obesity. The low-calorie diet did not significantly alter the microbial diversity and richness, or result in substantial shifts in the microbiota composition after successful weight loss. However, there were dramatic changes in the relative abundances of several microbiota, particularly reduced Proteobacteria and increased *Ruminococcus faeces* and *Bifidobacterium* sp. The reduction in Proteobacteria is likely advantageous because this group of Gram-negative bacteria is believed to play a role in proinflammatory conditions in the gut and pathogenesis of IBD [255,256]. Additionally, the elevated relative abundances of *Ruminococcus faeces* and *Bifidobacterium* sp. are also likely beneficial because these Gram-positive species are shown to be associated with reduced severity and fibrogenesis of non-alcoholic fatty liver disease [257] and colorectal cancer [258,259,260,261], respectively. 

Another clinical study, conducted by Dong et al. [262], examined a 16-week macronutrient standardised diet for weight loss for individuals with overweight or obesity. Microbiota results indicated no significant differences in microbiota richness and composition after weight loss. However, the group that successfully lost at least 5% of their body weight was associated with a reduction in the proportion of *Enterococcus*, while the group that did not have significant body weight loss had a significantly decreased abundance of *Klebsiella* and an increase of *Coprococcus* and *Collinsella*. Reduced *Klebsiella* is likely to be beneficial because it is associated with reduced risks of antimicrobial resistance [263,264,265] and pneumonia [266,267]. Increased *Coprococcus* is also likely to be favourable because its abundance is negatively correlated with depression [268,269] and Parkinson’s disease [270]. However, an enrichment of *Collinsella* is likely not favourable because it is associated with an increased risk of atherosclerosis [271], and type 2 diabetes [272]. 

Recently, a randomised controlled trial by Jian et al. [273] showed marked alterations in the gut microbiota composition of individuals with pre-diabetic overweight/obesity after weight loss. The 8-week low-energy diet resulted in significant increases in microbiota richness and alpha diversity, as well as Bray–Curtis values, and a significant reduction in the Firmicutes:Bacteroidetes ratio and capacity for butyrate production in the gut microbiota. Moreover, this weight loss was also associated with significantly reduced *Pseudobutyrivibrio* and *Bifidobacterium* and enriched *Akkermansia*. A reduced abundance of *Pseudobutyrivibrio* is associated with an increased risk of psoriatic arthritis [274]. However, an increased abundance of *Bifidobacterium* is likely to be beneficial because it is known as a probiotic agent that can reduce the risk of colorectal carcinoma [258,259,260,261] and IBD [275,276,277,278,279]. Moreover, the increased abundance of *Akkermansia* is likely to be beneficial because it is an important mucin-utiliser bacterium [280] associated with enhanced gut barrier function [281,282], improved insulin resistance [283], reduced metabolic endotoxemia [284] and cardiometabolic risk factor [285], and has been considered a biomarker for longevity [286]. 

A summary of human studies exploring the effect of post-obesity weight loss on gut microbiota is listed in Table 2.

Collectively, current evidence from animal models indicates that targeting weight loss via dietary interventions is associated with significant gut microbiota alterations and improves adipogenesis, fat deposition, and susceptibility to weight regain. Moreover, while there has not been any human study exploring the effects of yoyo dieting and gut microbiome, studies have shown that individuals with obesity after weight loss have significantly different microbiota signatures compared to those during the obesity state (Figure 1). Although human studies are important, the complexity of confounding factors in human data, especially from diet and lifestyle studies, often presents constant challenges. Animal models, on the other hand, offer a valuable research model that provides clear mechanistic insights into the intricate interactions between the gut microbiome and host physiology within a controlled experimental design. However, it is worth noting that there are differences between animal and human models [287], especially between mouse and human gut microbiota signatures [288]. This remains a key limitation in translating promising results from animal models to humans, particularly if these microbiome changes have host-specific physiological dependence. 

Given the relationship between gut health, obesity, and its comorbidities, modulating the gut microbiota emerges to be a promising strategy for managing weight and improving human health. However, it remains to be determined whether changes in gut microbiota directly or indirectly influence or are influenced by weight loss and the mechanisms behind repeated bouts of weight regain after loss. This highlights the importance of elucidating this complex relationship which remains underexplored in the existing literature. Future longitudinal studies, especially in humans, are needed to understand the potentially distinct effects of (1) continuous on–off diets, (2) weight fluctuations (the two components of yoyo dieting) and (3) gut-derived peptide hormonal imbalance on the gut microbiome. This will provide valuable insights into how to better manage weight regain susceptibility via gut microbiome modulation which can potentially pave the way to ameliorating current adverse metabolic issues and complications related to obesity.

## 5. Conclusions

There is emerging evidence indicating a relationship between yoyo dieting and gut microbiota. Current evidence suggests that an altered gut microbiome profile occurs with weight gain and persists long-term, even after successful weight loss. This might contribute to the greater susceptibility for weight relapse and potentially enhanced rate of weight gain after every cycle of weight loss and regain. Weight loss may be beneficial in ameliorating dysbiosis and intestinal inflammation. However, a full picture regarding yoyo dieting and gut health, including changes in the microbiota metabolites, remains to be determined. Moreover, the crosstalk between yoyo dieting and the gut signifies vital impacts in local and systemic effects on overall human health. There is little known about the molecular mechanism of how the gut influences the yoyo effect, such as how gut-derived peptides are impacted and their potential suitability as targets to combat cyclic weight regain. Furthermore, how yoyo dieting and associated changes in the microbiome affect the gut barrier and the enteric nervous system (intrinsic nervous system in the gut) has also been largely overlooked. Therefore, further studies, especially on humans, are needed to assess changes in peripheral peptides, epithelial lining, intestinal permeability, enteric neurons, and microbiome metabolites to crucially elucidate the effects of yoyo dieting on gut health and further explore the gastrointestinal system as a promising target for long-term weight maintenance.

## Figures and Tables

**Figure 1 nutrients-16-03170-f001:**
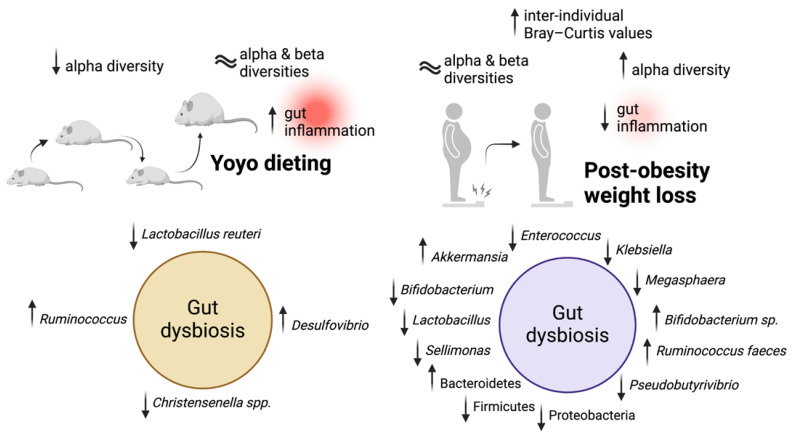
Overview of effects of yoyo dieting and post-obesity weight loss on gut health. Yoyo dieting refers to repeated phases of dieting and non-dieting leading to cyclic weight loss and regain. There has been growing evidence from animal studies suggesting that yoyo dieting is associated with increased susceptibility to weight regain, gut dysbiosis, and gut inflammation. The current literature also suggests that diet-induced weight loss is a potential approach to reduce gut inflammation. However, post-obesity weight loss in humans has also been associated with increased gut dysbiosis. ↑: increase; ↓: decrease.

**Table 1 nutrients-16-03170-t001:** Key animal studies exploring how yoyo dieting and weight loss interventions during yoyo dieting affect the gut microbiome. ↑: increase; ↓: decrease.

Overall Aims	Model	Diet Regime	Microbiota Changes and Metabolic Outcomes	Reference
Yoyo dieting changes microbiota composition and the administration of flavonoids might be beneficial in reduced weight regain	Male C57BL/6 mice	HFD → LFD → HFD → LFD	Yoyo mice: Different microbiota composition compared to mice fed a control diet or an HFD. ↓ alpha diversity compared to control mice. ↓ relative abundance of *Christensenella* spp. and *Lactobacillus reuteri* during obesity and after obesity. Yoyo mice treated with flavonoids: No effect on gut microbiota composition ↓ weight regain compared to yoyo mice. ↑ expression of UCP1 in brown adipose tissue.	[61]
The effect of yoyo dieting and the use of daisaikoto	Female C57BL/6 mice	Yoyo dieting only: HFD → LFD → HFD Yoyo dieting with daisaikoto supplementation: HFD → LFD + daisaikoto → HFD	Yoyo mice: ↓ alpha diversity compared to control mice. Similar relative abundances of Bacteroidetes and Firmicutes compared to control mice. ↑ relative abundance of *Rminococcus* and *Desulfovibrio* compared to control mice. Yoyo mice fed daisaikoto: ↓ alpha diversity compared to control mice. ↑ alpha diversity compared to yoyo mice. ↑ relative abundance of Bacteroidetes and ↓ relative abundance of Firmicutes compared to yoyo mice. Similar relative abundance of *Ruminococcus* and *Desulfovibrio* compared to control mice. ↓ relative abundance of *Ruminococcus* and *Desulfovibrio* compared to yoyo mice. ↓ body weight, visceral fat, and weight regain compared to yoyo mice.	[196]
Long term effect of yoyo dieting on faecal microbiota	Male C57BL/6 mice	(1) HFD → LFD → HFD → LFD → HFD → LFD → HFD → LFD → HFD → LFD or (2) LFD → HFD → LFD → HFD → LFD → HFD → LFD → HFD → LFD → HFD	Yoyo mice a fed a yoyo diet ending with a LFD: No significant differences in alpha and beta diversity compared to control mice. ↑ relative abundance of Bacteroidetes and *Lactobacillus* compared to obese mice and yoyo mice ending with a LFD. ↓ body weight at the end of the dietary intervention compared to yoyo mice fed a LFD/HFD cyclic diet. Yoyo mice fed a yoyo diet ending with a HFD: No significant differences in alpha and beta diversity compared to obese mice. ↑ relative abundance of Firmicutes (classes Erysipelotrichia and Bacilli) compared to control mice and yoyo mice ending with a LFD. ↑ body weight at the end of the dietary intervention compared to yoyo mice fed a HFD/LFD cyclic diet.	[62]

**Table 2 nutrients-16-03170-t002:** Key human studies exploring how post-obesity weight loss affects the gut microbiome. ↑: increase; ↓: decrease.

Overall Aims	Model	Diet Regime	Microbiota Changes	Reference
Examining effects of one-month caloric restriction on gut permeability and microbiome in female individuals with obesity	Female individuals with obesity	1-month low-calorie diet	No change in alpha and beta diversities compared to controls. ↓ relative abundance of Proteobacteria. ↑ relative abundance of *Ruminococcus faeces* and *Bifidobacterium* sp.	[114]
The intestinal microbiome predicts weight loss on a calorie-restricted diet in overweight and obesity individuals	Male and female individuals with overweight/obesity	16-week macronutrient standardised diet for weight loss	No gut microbiota difference at baseline. Individuals successfully lost at least 5% body weight: ↓ relative abundance of *Enterococcus.* ↓ relative abundance of *Klebsiella, Megasphaera, Sellimonas,* and *Lactobacillus* compared to those lost <5% body weight. Individuals lost <5% body weight: ↓ relative abundance of *Klebsiella.* ↑ relative abundance of *Coprococcus* and *Collinsella.*	[262]
A PREVIEW intervention study investigating the effect of low-energy diets on gut microbiome of individuals with overweight, or obesity and prediabetes	Male and female individuals with overweight/obesity and prediabetes	8-week low-energy diet	Individuals successfully lost at least 8% body weight ↑ microbiota richness and alpha diversity. ↑ inter-individual Bray-Curtis values. ↓ Firmicutes:Bacteroidetes ratio. ↓ relative abundance of *Pseudobutyrivibrio* and *Bifidobacterium*. ↑ relative abundance of *Akkermansia.*	[273]

## Data Availability

No new data were created or analysed in this study. Data sharing is not applicable to this article.

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
