# Peer review of "Yoyo Dieting, Post-Obesity Weight Loss, and Their Relationship with Gut Health"

_nutrients, 2024, doi:10.3390/nu16183170_

Round 1

Reviewer 1 Report

Comments and Suggestions for Authors

The paper “Yoyo dieting, post-obesity weight loss, and their relationship 2

with gut health” e summarise current evidence of the crosstalk between the gastrointestinal system and energy balance, and the effects of yoyo dieting on gut inflammation and gut microbiota reshaping. The review is meaningful, but there are some questions that need to be further improved or explained.

Comments:

Q1. Line 70-80, Please supplement the types of intestinal bacteria in obese and thin mice, so that readers can more clearly distinguish the correlation between intestinal flora diversity and animal fatness.

Q2. Line 109, The author mentioned that intestinal peptides are hormones, is there a well-established statement that peptides can be directly involved in regulating body functions as hormones? Or do polypeptides exert their functions indirectly by regulating other identified hormones?

Q3. Line 120-130, The authors refer to many tripeptides as peptide hormones secreted by the gut to explain their related functions. It is undeniable that many tripeptides have been found to have a variety of biological activities. However, is there any evidence to prove that these tripeptides are directly secreted by the gut to regulate physiological functions, or are they the products of intestinal microorganisms degrading the proteins ingested by the body?

Q4. Line 206-215, “there has not been 210 any indication that the currently approved medications are not suitable for use in the long term”. What kind of medicine is not suitable for long-term use? According to the authors, current weight loss drugs have shown some side effects and gastrointestinal symptoms, but are they still suitable for long-term use?

Q5. In the whole manuscript, the authors cited relevant papers appropriately and expressed them reasonably, but lacked their own summaries, which was suggested to be supplemented at the end of each section.

Q6. It is highly recommended that the authors incorporate a diagram illustrating the intestinal health status during the process of obesity, weight-loss and relapse, thereby enhancing the manuscript's overall qualities.

Author Response

Dear reviewer,

Thank you for giving us the opportunity to submit a revised draft of our manuscript titled ‘Yoyo dieting, post-obesity weight loss, and their relationship with gut health’ to Nutrients. We appreciate the time and effort that you have dedicated to providing valuable and insightful feedback on our manuscript. We have addressed your comments (please see below) and we believe that the manuscript is now improved. Sentences highlighted in blue are responses addressing the reviewer’s comments and track changes that have also been made within the manuscript.

Q1. Line 70-80, Please supplement the types of intestinal bacteria in obese and thin mice, so that readers can more clearly distinguish the correlation between intestinal flora diversity and animal fatness.

Thank you for your feedback. We have added additional information on the potential gut microbiota changes that lead to gut dysbiosis associated with obesity. This was addressed by adding “(e.g. enriched Fusobacterium and reduced Oscillospira, Ruminococcaceae, Coriobacteriaceae, Odoribacter splanchnicus, Akkermansia muciniphila and Bifidobacterium longum)” to lines 75–77, “compared to controls (e.g. enriched Akkermnansia muciniphila, Alistipes indistinctus, Odoribacter splanchnicus, Clostridium sp. CAG:413, and Bifidobacterium longum)” to lines 79–81. We have also added “(enriched Bacteroides uniformisParabacteroides merdae, Alistipes putredinis, Eubacterium rectale and reduced Bacteroides thetaiotaomicron) to lines 86–87.

Q2. Line 109, The author mentioned that intestinal peptides are hormones, is there a well-established statement that peptides can be directly involved in regulating body functions as hormones? Or do polypeptides exert their functions indirectly by regulating other identified hormones?

It is well established that gastrointestinal hormones are peptides, and the gastrointestinal tract is the largest endocrine organ in the body that produces peptide hormones. Enteroendocrine cells in the gastrointestinal tract release hormones that can act locally, on other cells (including immune cells), nerve endings, or organs at remote sites including the CNS. The senior author of this paper has published a comprehensive review discussing the role of the gut as a sensory organ (Furness et al. 2013). Gut peptides also regulate energy balance which has been discussed in the manuscript, including PYY, GLP-1, CKK, OXM, and ghrelin, which are known to regulate feeding behaviours (Tang-Christensen, Vrang, and Larsen 2001) and were discussed in detail in this section (lines 123–137). The name ‘peptide hormones’ has been used for these mentioned gut peptides in the literature (PYY (Karra, Chandarana, and Batterham 2009; Batterham and Bloom 2003), GLP-1 (Holst 2007; Paternoster and Falasca 2018), CKK (Rehfeld et al. 2007; de Souza et al. 2020), OXM (Pocai 2014; Ouberai et al. 2017), Ghrelin (Wang et al. 2021; Mathur et al. 2021)). Moreover, other mentioned peptides secreted by adipose tissue and pancreas, which also play key roles acting via neuroendocrine mechanisms to convey information on energy status from the periphery to the brain hence affecting energy homeostasis and appetite control (discussed in lines 144–160), are also often referred as peptide hormones in the literature (Leptin (Kaneko et al. 2022; Obradovic et al. 2021), Adiponectin (Tworoger, Mantzoros, and Hankinson 2007), Insulin (Wilcox 2005), Amylin (Ludvik et al. 1997; Bower et al. 2018), PP (Banerjee and Onyuksel 2012), GIP (Zhao et al. 2021)). We also previously mentioned that these peptide hormones regulate energy balance by directly or indirectly controlling the activity of orexin and anorexic neurons within integrating brain regions such as the hypothalamus (Méquinion, Foldi, and Andrews 2019) (lines 120–123). To add more clarity to this section, we have replaced all instances of “peptide” (which stands alone) with “peptide hormone” as appropriate throughout section 2.1.

Q3. Line 120-130, The authors refer to many tripeptides as peptide hormones secreted by the gut to explain their related functions. It is undeniable that many tripeptides have been found to have a variety of biological activities. However, is there any evidence to prove that these tripeptides are directly secreted by the gut to regulate physiological functions, or are they the products of intestinal microorganisms degrading the proteins ingested by the body?

Yes, there are approximately 12 major enteroendocrine cell types that have been identified, which collectively secrete more than 20 hormones (including those mentioned in the manuscript such as PYY, GLP-1, CCK, OXM and ghrelin) that regulate physiological functions (De Silva and Bloom 2012; Xiao, Jin, and Zhang 2024; Habib et al. 2013; Baggio and Drucker 2007; Liddle 1997; Buffa, Solcia, and Go 1976; Posovszky and Wabitsch 2014; Cohen et al. 2003; Date et al. 2000; Ibrahim Abdalla 2015).

Q4. Line 206-215, “there has not been any indication that the currently approved medications are not suitable for use in the long term”. What kind of medicine is not suitable for long-term use? According to the authors, current weight loss drugs have shown some side effects and gastrointestinal symptoms, but are they still suitable for long-term use?

The currently marketed anti-obesity drugs targeting the peripheral mechanisms, specifically Liraglutide, Semaglutide, Orlistat, and Tirzepatide (newly updated to the manuscript) have side effects but are still approved for long-term usage (Jeong and Priefer 2022; Tak and Lee 2021). There are two FDA-approved anti-obesity drugs that have been approved for short-term usage only (up to 3 months), including Phentermine and Diethylpropion which are both sympathomimetic agents to suppress appetite (Tchang et al. 2024) and do not act on the peripheral mechanism, hence are not the focus of this section 2.1 and this review. However, we acknowledge that this sentence requires rewording to deliver a clearer message. To tackle this, we have replaced “To date, there has not been any indication that the currently approved medications are not suitable for use in the long term” with “To date, these medications are approved for long-term usage” (lines 245–246). We have also updated the manuscript with the 4th approved anti-obesity drug, Tirzepatide which is a dual GIP and GLP–1 receptor agonist acting as an appetite suppressant, to the list of new weight loss therapeutics (lines 241–242).

Q5. In the whole manuscript, the authors cited relevant papers appropriately and expressed them reasonably, but lacked their own summaries, which was suggested to be supplemented at the end of each section.

Thank you for your suggestion. A summary paragraph has mostly been written at the end of every section throughout the manuscript ((lines 65-70; lines 93-104; lines 248-249; lines 280-283; lines 319-321; lines 355-363; lines 441-445). However, we acknowledge that a summary paragraph for section 4.2 was missing and therefore have added two new paragraphs at the end of this section reviewing the strengths and limitations of animal and human study models reviewed in this manuscript (lines 549-563) and discussing knowledge gaps and future directions (lines 571-583).

Q6. It is highly recommended that the authors incorporate a diagram illustrating the intestinal health status during the process of obesity, weight-loss and relapse, thereby enhancing the manuscript's overall qualities.

Thank you for your suggestion. We have added a figure at the end of section 4 to illustrate the current knowledge regarding obesity and yoyo dieting on gut health (lines 564-570).

Thank you once again for your feedback on how to improve this manuscript. We look forward to hearing from you in due time regarding our submission and to responding to any further questions and comments you may have.

Sincerely,

Kate Phuong-Nguyen, Sean McGee, Kathryn Aston-Mourney, Bryony Mcneill, Malik Mahmood and Leni Rivera

References

Baggio, Laurie L., and Daniel J. Drucker. 2007. 'Biology of Incretins: GLP-1 and GIP', Gastroenterology, 132: 2131-57.

Banerjee, A., and H. Onyuksel. 2012. 'Human pancreatic polypeptide in a phospholipid-based micellar formulation', Pharm Res, 29: 1698-711.

Batterham, R. L., and S. R. Bloom. 2003. 'The gut hormone peptide YY regulates appetite', Ann N Y Acad Sci, 994: 162-8.

Bower, Rebekah L., Lauren Yule, Tayla A. Rees, Giuseppe Deganutti, Erica R. Hendrikse, Paul W. R. Harris, Renata Kowalczyk, Zachary Ridgway, Amy G. Wong, Katarzyna Swierkula, Daniel P. Raleigh, Augen A. Pioszak, Margaret A. Brimble, Christopher A. Reynolds, Christopher S. Walker, and Debbie L. Hay. 2018. 'Molecular Signature for Receptor Engagement in the Metabolic Peptide Hormone Amylin', ACS Pharmacology & Translational Science, 1: 32-49.

Buffa, Roberto, Enrico Solcia, and Vay Liang W Go. 1976. 'Immunohistochemical identification of the cholecystokinin cell in the intestinal mucosa', Gastroenterology, 70: 528-32.

Cohen, Mark A., Sandra M. Ellis, Carel W. Le Roux, Rachel L. Batterham, Adrian Park, Michael Patterson, Gary S. Frost, Mohammad A. Ghatei, and Stephen R. Bloom. 2003. 'Oxyntomodulin Suppresses Appetite and Reduces Food Intake in Humans', The Journal of Clinical Endocrinology & Metabolism, 88: 4696-701.

Date, Yukari, Masayasu Kojima, Hiroshi Hosoda, Akira Sawaguchi, Muhtashan S. Mondal, Tatsuo Suganuma, Shigeru Matsukura, Kenji Kangawa, and Masamitsu Nakazato. 2000. 'Ghrelin, a Novel Growth Hormone-Releasing Acylated Peptide, Is Synthesized in a Distinct Endocrine Cell Type in the Gastrointestinal Tracts of Rats and Humans**This work was supported in part by grants-in-aid from the Ministry of Education, Science, Sports, and Culture, Japan, and the Ministry of Health and Welfare, Japan (to M.N.)', Endocrinology, 141: 4255-61.

De Silva, A., and S. R. Bloom. 2012. 'Gut Hormones and Appetite Control: A Focus on PYY and GLP-1 as Therapeutic Targets in Obesity', Gut Liver, 6: 10-20.

de Souza, Arnaldo Henrique, Jiayin Tang, Amanjot Kaur Yadev, Samuel T. Saghafi, Carly R. Kibbe, Amelia K. Linnemann, Matthew J. Merrins, and Dawn Belt Davis. 2020. 'Intra-islet GLP-1, but not CCK, is necessary for β-cell function in mouse and human islets', Scientific Reports, 10: 2823.

Furness, John B., Leni R. Rivera, Hyun-Jung Cho, David M. Bravo, and Brid Callaghan. 2013. 'The gut as a sensory organ', Nature Reviews Gastroenterology & Hepatology, 10: 729-40.

Habib, A. M., P. Richards, G. J. Rogers, F. Reimann, and F. M. Gribble. 2013. 'Co-localisation and secretion of glucagon-like peptide 1 and peptide YY from primary cultured human L cells', Diabetologia, 56: 1413-6.

Holst, J. J. 2007. 'The physiology of glucagon-like peptide 1', Physiol Rev, 87: 1409-39.

Ibrahim Abdalla, M. M. 2015. 'Ghrelin - Physiological Functions and Regulation', Eur Endocrinol, 11: 90-95.

Jeong, Dagam, and Ronny Priefer. 2022. 'Anti-obesity weight loss medications: Short-term and long-term use', Life Sciences, 306: 120825.

Kaneko, Kentaro, Yukihiro Takekuma, Tsuyoshi Goto, and Kousaku Ohinata. 2022. 'An orally active plant Rubisco-derived peptide increases neuronal leptin responsiveness', Scientific Reports, 12: 8599.

Karra, E., K. Chandarana, and R. L. Batterham. 2009. 'The role of peptide YY in appetite regulation and obesity', J Physiol, 587: 19-25.

Liddle, R. A. 1997. 'Cholecystokinin cells', Annu Rev Physiol, 59: 221-42.

Ludvik, B., A. Kautzky-Willer, R. Prager, K. Thomaseth, and G. Pacini. 1997. 'Amylin: history and overview', Diabet Med, 14 Suppl 2: S9-13.

Mathur, Nimisha, Syed F. Mehdi, Manasa Anipindi, Monowar Aziz, Sawleha A. Khan, Hema Kondakindi, Barbara Lowell, Ping Wang, and Jesse Roth. 2021. 'Ghrelin as an Anti-Sepsis Peptide: Review', Frontiers in Immunology, 11.

Méquinion, M., C. J. Foldi, and Z. B. Andrews. 2019. 'The Ghrelin-AgRP Neuron Nexus in Anorexia Nervosa: Implications for Metabolic and Behavioral Adaptations', Front Nutr, 6: 190.

Obradovic, Milan, Emina Sudar-Milovanovic, Sanja Soskic, Magbubah Essack, Swati Arya, Alan J. Stewart, Takashi Gojobori, and Esma R. Isenovic. 2021. 'Leptin and Obesity: Role and Clinical Implication', Frontiers in Endocrinology, 12.

Ouberai, Myriam M., Ana L. Gomes Dos Santos, Sonja Kinna, Shimona Madalli, David C. Hornigold, David Baker, Jacqueline Naylor, Laura Sheldrake, Dominic J. Corkill, John Hood, Paolo Vicini, Shahid Uddin, Steven Bishop, Paul G. Varley, and Mark E. Welland. 2017. 'Controlling the bioactivity of a peptide hormone in vivo by reversible self-assembly', Nature communications, 8: 1026.

Paternoster, Silvano, and Marco Falasca. 2018. 'Dissecting the Physiology and Pathophysiology of Glucagon-Like Peptide-1', Frontiers in Endocrinology, 9.

Pocai, A. 2014. 'Action and therapeutic potential of oxyntomodulin', Mol Metab, 3: 241-51.

Posovszky, Carsten, and Martin Wabitsch. 2014. 'Regulation of Appetite, Satiation, and Body Weight by Enteroendocrine Cells. Part 1: Characteristics of Enteroendocrine Cells and Their Capability of Weight Regulation', Hormone Research in Paediatrics, 83: 1-10.

Rehfeld, J. F., L. Friis-Hansen, J. P. Goetze, and T. V. Hansen. 2007. 'The biology of cholecystokinin and gastrin peptides', Curr Top Med Chem, 7: 1154-65.

Rinninella, Emanuele, Pauline Raoul, Marco Cintoni, Francesco Franceschi, Giacinto Abele Donato Miggiano, Antonio Gasbarrini, and Maria Cristina Mele. 2019. 'What is the healthy gut microbiota composition? A changing ecosystem across age, environment, diet, and diseases', Microorganisms, 7: 14.

Tak, Y. J., and S. Y. Lee. 2021. 'Long-Term Efficacy and Safety of Anti-Obesity Treatment: Where Do We Stand?', Curr Obes Rep, 10: 14-30.

Tang-Christensen, M, N Vrang, and PJ Larsen. 2001. 'Glucagon-like peptide containing pathways in the regulation of feeding behaviour', International Journal of Obesity, 25: S42-S47.

Tchang, BG., M. Aras, RB. Kumar, and LJ. Aronne. 2024. 'Pharmacologic Treatment of Overweight and Obesity in Adults', Endotext [Internet], Accessed 9th September.

Tworoger, Shelley S., Christos Mantzoros, and Susan E. Hankinson. 2007. 'Relationship of Plasma Adiponectin With Sex Hormone and Insulin-like Growth Factor Levels', Obesity, 15: 2217-24.

Wang, Yue, Shimeng Guo, Youwen Zhuang, Ying Yun, Peiyu Xu, Xinheng He, Jia Guo, Wanchao Yin, H. Eric Xu, Xin Xie, and Yi Jiang. 2021. 'Molecular recognition of an acyl-peptide hormone and activation of ghrelin receptor', Nature communications, 12: 5064.

Wilcox, G. 2005. 'Insulin and insulin resistance', Clin Biochem Rev, 26: 19-39.

Xiao, Yuchen, Lingjing Jin, and Chao Zhang. 2024. 'From a hunger-regulating hormone to an antimicrobial peptide: gastrointestinal derived circulating endocrine hormone-peptide YY exerts exocrine antimicrobial effects against selective gut microbiota', Gut Microbes, 16: 2316927.

Zhao, Fenghui, Chao Zhang, Qingtong Zhou, Kaini Hang, Xinyu Zou, Yan Chen, Fan Wu, Qidi Rao, Antao Dai, Wanchao Yin, Dan-Dan Shen, Yan Zhang, Tian Xia, Raymond C. Stevens, H. Eric Xu, Dehua Yang, Lihua Zhao, and Ming-Wei Wang. 2021. 'Structural insights into hormone recognition by the human glucose-dependent insulinotropic polypeptide receptor', Elife, 10: e68719.

Reviewer 2 Report

Comments and Suggestions for Authors

While the review “Yoyo dieting, post-obesity weight loss, and their relationship with gut health” covers a broad range of topics, the focus sometimes shifts away from the main topic. For example, the discussion on the role of gut hormones and peptides is extensive but could be more tightly linked to yoyo dieting. The authors should ensure that all sections directly contribute to understanding the impact of yoyo dieting on gut health.

·      The manuscript does not clearly explain how studies were selected for inclusion in the review. A section detailing the literature search strategy, including databases used, keywords, inclusion/exclusion criteria, and how biases were mitigated, would strengthen the methodological rigor.

·      The review mentions several factors, such as dietary composition and physical activity, that could influence the relationship between yoyo dieting and gut health. However, these factors are not thoroughly analyzed or controlled across the cited studies. A more critical examination of how these confounding variables might affect the outcomes would provide a deeper understanding of the topic.

·      A significant portion of the evidence presented in the manuscript is derived from animal studies. While these studies are valuable for understanding basic mechanisms, the authors should more critically discuss the limitations of translating these findings to humans. The review could benefit from a more balanced inclusion of human studies and a discussion of how differences between species may affect the applicability of the results.

·      Some sections of the manuscript present data inconsistently, making it difficult for readers to compare findings across studies. For example, the reporting of microbiota changes could be standardized, perhaps using summary tables that highlight key findings such as changes in alpha diversity, specific bacterial taxa, and associated metabolic outcomes.

·      The manuscript would benefit from additional figures or diagrams, especially those that could illustrate complex mechanisms, such as the role of gut peptides in weight regain or the impact of yoyo dieting on gut microbiota diversity. Visual aids would enhance the reader's understanding and engagement.

·      The existing table is a good start, but it could be expanded to include human studies and to compare findings across different types of interventions (e.g., dietary changes, bariatric surgery). This would provide a more comprehensive overview of the research landscape.

·      While the review is comprehensive, it does not offer new hypotheses, models, or frameworks that could drive the field forward. The authors could enhance the novelty of the manuscript by proposing new research directions, identifying novel mechanisms, or suggesting innovative interventions based on the findings discussed.

·      References are not according to journal format, revise them.

Overall, I recommend authors to revise their manuscript considering the above mentioned comments.

Author Response

Dear reviewer,

Thank you for giving us the opportunity to submit a revised draft of our manuscript titled ‘Yoyo dieting, post-obesity weight loss, and their relationship with gut health’ to Nutrients. We appreciate the time and effort that you have dedicated to providing valuable and insightful feedback on our manuscript. We have addressed your comments (please see below) and we believe that the manuscript is now improved. Sentences highlighted in blue are responses addressing the reviewer’s comments and track changes that have also been made within the manuscript.

  • The manuscript does not clearly explain how studies were selected for inclusion in the review. A section detailing the literature search strategy, including databases used, keywords, inclusion/exclusion criteria, and how biases were mitigated, would strengthen the methodological rigor.

Thank you for your comment and suggestion. The methodology for this review has not been included as this is a narrative review rather than a systematic review. We performed a comprehensive search of databases such as PubMed and Google Scholar to identify relevant studies. Keywords including 'obesity', ‘yoyo dieting', 'weight cycling', 'weight regain', 'gut microbiome', 'gut dysbiosis', 'gut inflammation', 'weight loss', 'peripheral peptides', 'gut peptide', ‘gut hormone’, ‘energy balance’, ‘energy homeostasis’,  ‘dietary intervention’, ‘calorie restriction’, and ‘low energy diet’ were employed during the search process.

  • The review mentions several factors, such as dietary composition and physical activity, that could influence the relationship between yoyo dieting and gut health. However, these factors are not thoroughly analyzed or controlled across the cited studies. A more critical examination of how these confounding variables might affect the outcomes would provide a deeper understanding of the topic.

Thank you for your suggestion. As highlighted in this narrative review, there are only a small number of studies that have investigated the relationship between the gut microbiome and yoyo dieting (mainly animal studies), and post-obesity weight loss (mainly human studies), highlighting the need for more investigation in this area (lines 549-555). While it is established that diets, diseases, medications, and environmental factors are key factors influencing gut microbiota changes, what is currently known are mostly associations rather than causations (lines 571-577). Therefore, it remains a great challenge to critically analyse the confounding factors (obesity, energy balance, gut inflammation, dietary intervention, microbiota changes) affecting weight loss and weight regain (lines 555-557). This is also why we have broadened this review to include the effects of yoyo dieting on gut inflammation, gut peptides and post-obesity weight loss for a comprehensive review of this topic. This has been addressed by adding 2 new paragraphs at the end of section 4.2.2 (lines 549-563; lines 571-583).

  • A significant portion of the evidence presented in the manuscript is derived from animal studies. While these studies are valuable for understanding basic mechanisms, the authors should more critically discuss the limitations of translating these findings to humans. The review could benefit from a more balanced inclusion of human studies and a discussion of how differences between species may affect the applicability of the results.

To date, all of the studies around yoyo dieting and gut health have come from animal studies, and human studies are lacking. We acknowledge that there are differences between animal and human models and there are limitations with translating these findings to humans. To address these limitations, we have added 2 short paragraphs at the end of section 3.1 (lines 319-321) and section 4.2.2 (lines 549-563).

  • Some sections of the manuscript present data inconsistently, making it difficult for readers to compare findings across studies. For example, the reporting of microbiota changes could be standardized, perhaps using summary tables that highlight key findings such as changes in alpha diversity, specific bacterial taxa, and associated metabolic outcomes.

Thank you for your suggestion. We have amended this accordingly in Table 1 (lines 439-440).

  • The manuscript would benefit from additional figures or diagrams, especially those that could illustrate complex mechanisms, such as the role of gut peptides in weight regain or the impact of yoyo dieting on gut microbiota diversity. Visual aids would enhance the reader's understanding and engagement.

Thank you for your suggestion. We have added a figure at the end of section 4 to illustrate the current knowledge regarding obesity and yoyo dieting on gut health (lines 564-570).

  • The existing table is a good start, but it could be expanded to include human studies and to compare findings across different types of interventions (e.g., dietary changes, bariatric surgery). This would provide a more comprehensive overview of the research landscape.

Thank you for your suggestion. The literature on the effects of dietary changes, bariatric surgery etc. on the gut microbiome is extensive, and many excellent reviews discuss these (Ciobârcă et al. 2020; Georgiou et al. 2022; Akagbosu et al. 2022). However, we acknowledge the importance of including human studies on the table. We performed a comprehensive literature search and found that there are no human studies that have explored the effect of yoyo dieting on the gut microbiome, and only 3 human studies examining the effects of post-obesity weight loss on the gut microbiome as listed in section 4.2.2. Therefore, we have added a new table Table 2 (lines 544-546) summarising the evidence of post-obesity weight loss on the gut microbiome from human studies which is in line with the focus of this review (yoyo dieting, post-obesity weight loss, and the gut microbiome).

  • While the review is comprehensive, it does not offer new hypotheses, models, or frameworks that could drive the field forward. The authors could enhance the novelty of the manuscript by proposing new research directions, identifying novel mechanisms, or suggesting innovative interventions based on the findings discussed.

Thank you for your feedback.  We have added a short paragraph summarising the knowledge gaps and future directions at the end of section 4 (lines 571–583).

  • References are not according to journal format, revise them.

This has been amended.

Thank you once again for your feedback on how to improve this manuscript. We look forward to hearing from you in due time regarding our submission and to responding to any further questions and comments you may have.

Sincerely,

Kate Phuong-Nguyen, Sean McGee, Kathryn Aston-Mourney, Bryony Mcneill, Malik Mahmood and Leni Rivera

References

Akagbosu, Cynthia Omoge, Evan Paul Nadler, Shira Levy, and Suchitra Kaveri Hourigan. 2022. 'The Role of the Gut Microbiome in Pediatric Obesity and Bariatric Surgery', International journal of molecular sciences, 23: 15421.

Ciobârcă, Daniela, Adriana Florinela Cătoi, Cătălin Copăescu, Doina Miere, and Gianina CriÈ™an. 2020. 'Bariatric Surgery in Obesity: Effects on Gut Microbiota and Micronutrient Status', Nutrients, 12: 235.

Georgiou, K., N. A. Belev, T. Koutouratsas, H. Katifelis, and M. Gazouli. 2022. 'Gut microbiome: Linking together obesity, bariatric surgery and associated clinical outcomes under a single focus', World J Gastrointest Pathophysiol, 13: 59-72.

Reviewer 3 Report

Comments and Suggestions for Authors

The article provides a comprehensive review of the current understanding of yoyo dieting, post-obesity weight loss, and their intricate relationship with gut health. Overall the article is thorough in its coverage of the literature, it lacks critical analysis of the studies reviewed. For instance, the discussion on gut peptides and their role in weight regain could benefit from a more critical assessment of the methodologies used in the studies cited. Additionally, the article could discuss potential biases or limitations in the research more explicitly.

Please avoid using the first person in the article

The abstract summarizes the article but fails to highlight specific findings or knowledge gaps. For example, it mentions that the review "identifies current knowledge gaps within the field" but doesn't specify what those gaps are. The abstract would benefit from a more explicit mention of the key conclusions and their significance.

The introduction, while informative, does not present a clear thesis statement or central argument that guides the reader through the article. A more explicit statement of the article's purpose would help to focus the review and make it easier for readers to follow the main points.

Section 2. The Influence of the Gut in Weight Regain covers the role of gut peptides in energy balance, but doesn’t consistently tie these findings back to the topic of yoyo dieting. For instance, the discussion on gut peptides like GLP-1 and ghrelin is informative, but the connection to how these peptides influence weight cycling is not clearly made – please add some explications about how these peptides work on the weight cycling  and focus on key details

In section 3. Yoyo Dieting and Gut Inflammation draws heavily from animal studies to discuss gut inflammation, which limits the applicability of the findings to humans. For example, the text discusses how "mice fed a yoyo diet showed significantly lower C1q/tumour necrosis factor-related protein-3 (CTRP3) expression," but it doesn’t sufficiently address the limitations of translating these findings to human physiology. – please add the limitations of these findings - A more critical analysis of the exemplified studies would be helpful.

Provide a more in-depth critique of the studies reviewed, focusing on potential biases, limitations, and the variability of results across different models and species. This will add depth to your discussion and help identify areas where further research is needed.

Same for section 4. Yoyo Dieting and Gut Microbiome Lack of Critical Analysis which presents findings on how yoyo dieting affects the gut microbiome, it doesn’t critically assess the limitations of these studies. For example, the authors notes that "mice fed a yoyo diet had significantly lower alpha diversity," but it doesn’t discuss potential biases in these studies or the variability in microbiome responses across different species.

The discussion of microbiome changes is not well contextualized within the broader field. The article notes changes like the "increased relative abundance of Firmicutes," but it doesn’t compare these findings to other dietary patterns or interventions, leaving the reader with an incomplete understanding of the significance of these changes – please rewrite this part.

The conclusion summarizes the review but does not leave the reader with a strong, lasting impression. It mentions that "further studies are needed," but does not provide specific directions or innovative ideas for future research. The conclusion should be more concrete in its recommendations.

Author Response

Dear reviewer,

Thank you for giving us the opportunity to submit a revised draft of our manuscript titled ‘Yoyo dieting, post-obesity weight loss, and their relationship with gut health’ to Nutrients. We appreciate the time and effort that you have dedicated to providing valuable and insightful feedback on our manuscript. We have addressed your comments (please see below) and we believe that the manuscript is now improved. Sentences highlighted in blue are responses addressing the reviewer’s comments and track changes that have also been made within the manuscript.

  • The article provides a comprehensive review of the current understanding of yoyo dieting, post-obesity weight loss, and their intricate relationship with gut health. Overall the article is thorough in its coverage of the literature, it lacks critical analysis of the studies reviewed. For instance, the discussion on gut peptides and their role in weight regain could benefit from a more critical assessment of the methodologies used in the studies cited. Additionally, the article could discuss potential biases or limitations in the research more explicitly.

Thank you for your feedback. To address this, we have two short paragraphs discussing the limitations and knowledge gaps around gut peptides and yoyo dieting in section 2.1 (lines 137-142; lines 195-200). We have also added two paragraphs discussing limitations and future research at the end of section 4 (lines 535-549; lines 557-569).

  • Please avoid using the first person in the article

This has been amended throughout the manuscript.

  • The abstract summarizes the article but fails to highlight specific findings or knowledge gaps. For example, it mentions that the review "identifies current knowledge gaps within the field" but doesn't specify what those gaps are. The abstract would benefit from a more explicit mention of the key conclusions and their significance.

This has been amended by adding information on current knowledge gaps in the field of yoyo dieting and gut microbiome to the abstract (lines 23-25).

  • The introduction, while informative, does not present a clear thesis statement or central argument that guides the reader through the article. A more explicit statement of the article's purpose would help to focus the review and make it easier for readers to follow the main points.

Thank you for your feedback. At the end of section 1, we have discussed the purpose of the review with a sentence highlighting the increased prevalence of yoyo dieting warranting further investigation (lines 97-99) and two sentences summarising the focus of this review (lines 99-104).

  • Section 2. The Influence of the Gut in Weight Regain covers the role of gut peptides in energy balance, but doesn’t consistently tie these findings back to the topic of yoyo dieting. For instance, the discussion on gut peptides like GLP-1 and ghrelin is informative, but the connection to how these peptides influence weight cycling is not clearly made – please add some explications about how these peptides work on the weight cycling  and focus on key details

Thank you for your feedback. Currently, there are no studies that have investigated changes in gut peptides and yoyo dieting. However, we have discussed the changes in gut peptides in weight regain following weight loss given the importance of this topic, which is the focus of Section 2.

  • In section 3. Yoyo Dieting and Gut Inflammation draws heavily from animal studies to discuss gut inflammation, which limits the applicability of the findings to humans. For example, the text discusses how "mice fed a yoyo diet showed significantly lower C1q/tumour necrosis factor-related protein-3 (CTRP3) expression," but it doesn’t sufficiently address the limitations of translating these findings to human physiology. – please add the limitations of these findings - A more critical analysis of the exemplified studies would be helpful.

To date, all of the studies around yoyo dieting and gut health have come from animal studies, and human studies are lacking. We acknowledge that there are differences between animal and human models and there are limitations with translating these findings to humans. To address these limitations, we have added a short paragraph at the end of sections 3.1 (lines 305-307) and 4.2.2 (lines 535-549).

  • Provide a more in-depth critique of the studies reviewed, focusing on potential biases, limitations, and the variability of results across different models and species. This will add depth to your discussion and help identify areas where further research is needed.

This has been addressed by two new paragraphs at the end of section 4.2.2 discussing potential biases, limitations, and the variability of results across human and mouse models, especially on the gut microbiota difference between the two species (lines 535-549), and identified knowledge gap as well as suggesting future directions (lines 557-569).

  • Same for section 4. Yoyo Dieting and Gut Microbiome Lack of Critical Analysis which presents findings on how yoyo dieting affects the gut microbiome, it doesn’t critically assess the limitations of these studies. For example, the authors notes that "mice fed a yoyo diet had significantly lower alpha diversity," but it doesn’t discuss potential biases in these studies or the variability in microbiome responses across different species.

This has been amended by adding a short paragraph discussing the limitations of microbiome studies at the end of section 4.2.2 (lines 535-549).

  • The discussion of microbiome changes is not well contextualized within the broader field. The article notes changes like the "increased relative abundance of Firmicutes," but it doesn’t compare these findings to other dietary patterns or interventions, leaving the reader with an incomplete understanding of the significance of these changes – please rewrite this part.

It is well established that Firmicutes and Bacteroidetes are the most predominant microbial phyla in the human and mouse gut microbiota (Rinninella et al. 2019). Changes in the relative abundance of the two major phyla Firmicutes and Bacteroidetes and their families/genera/species have been seen as one of the main focused results in most gut microbiome studies. Although there are conflicting findings, a Firmicutes:Bacteroidetes ratio greater than 1 (more Firmicutes, less Bacteroidetes) is often associated with the obese phenotype, while a ratio lower than 1 (more Firmicutes, less Bacteroidetes) is associated with a lean phenotype (lines 465-467) (Ahmed et al. 2005; Ley et al. 2006; Armougom et al. 2009; Turnbaugh et al. 2009). In this narrative review manuscript, specifically in section 4 reviewing the relationship between yoyo dieting and gut microbiome, we focused on reporting significant changes in alpha diversity (distribution of microbiota richness and evenness), beta diversity (describe similarities and dissimilarities between diet treatment groups), and relative abundance of specific taxa, and concluded each paragraph with an explanation of the potential impacts/predictions of microbiome changes to general health (lines 386-395; lines 339-404; lines 415-422; lines 449-455; lines 463-472; lines 478-482; lines 492-498; lines 502-514; and lines 517-529). Moreover, to add more clarity to this microbiome section and help readers navigate information more easily, we have summarised key microbiome changes taken from animal studies (exploring the effects of yoyo dieting on the gut microbiome) in Table 1 (lines 425-426) and human studies (exploring the effects of post-obesity weight loss on the gut microbiome) in Table 2 (lines 530-532).

  • Some sections of the manuscript present data inconsistently, making it difficult for readers to compare findings across studies. For example, the reporting of microbiota changes could be standardized, perhaps using summary tables that highlight key findings such as changes in alpha diversity, specific bacterial taxa, and associated metabolic outcomes.

Thank you for your feedback. To address this, we have amended Table 1 (Key animal studies exploring how yoyo dieting and weight loss interventions during yoyo dieting affect the gut microbiome –lines 425-426) and added Table 2 (Key human studies exploring how post-obesity weight loss affects the gut microbiome – lines 530-532) to the manuscript.

  • The conclusion summarizes the review but does not leave the reader with a strong, lasting impression. It mentions that "further studies are needed," but does not provide specific directions or innovative ideas for future research. The conclusion should be more concrete in its recommendations.

In the conclusion section, future directions have been discussed in detail (lines 584-588) with clear objectives to target identified knowledge gaps in the literature (lines 576-584). Specifically, it would be valuable if future studies could assess changes in peripheral peptides, epithelial lining, intestinal permeability, enteric neurons, and microbiome metabolites to crucially elucidate the effects of yoyo dieting on gut health and further explore the gastrointestinal system as a promising target for long-term weight maintenance (lines 584-588). We also added extra information that more studies “especially on humans” (line 584) are warranted.

Thank you once again for your feedback on how to improve this manuscript. We look forward to hearing from you in due time regarding our submission and to responding to any further questions and comments you may have.

Yours Sincerely,

Kate Phuong-Nguyen, Sean McGee, Kathryn Aston-Mourney, Bryony Mcneill, Malik Mahmood and Leni Rivera

References

Ahmed, Naila, Bahar Mirshekar-Syahkal, Lauren Kennish, Nikolaos Karachalias, Roya Babaei-Jadidi, and Paul J. Thornalley. 2005. 'Assay of advanced glycation endproducts in selected beverages and food by liquid chromatography with tandem mass spectrometric detection', Molecular nutrition & food research, 49: 691-99.

Armougom, Fabrice, Mireille Henry, Bernard Vialettes, Denis Raccah, and Didier Raoult. 2009. 'Monitoring bacterial community of human gut microbiota reveals an increase in Lactobacillus in obese patients and Methanogens in anorexic patients', PloS one, 4: e7125.

Ley, Ruth E, Peter J Turnbaugh, Samuel Klein, and Jeffrey I Gordon. 2006. 'Human gut microbes associated with obesity', Nature, 444: 1022-23.

Turnbaugh, Peter J, Micah Hamady, Tanya Yatsunenko, Brandi L Cantarel, Alexis Duncan, Ruth E Ley, Mitchell L Sogin, William J Jones, Bruce A Roe, and Jason P Affourtit. 2009. 'A core gut microbiome in obese and lean twins', Nature, 457: 480-84.

Rinninella, Emanuele, Pauline Raoul, Marco Cintoni, Francesco Franceschi, Giacinto Abele Donato Miggiano, Antonio Gasbarrini, and Maria Cristina Mele. 2019. 'What is the healthy gut microbiota composition? A changing ecosystem across age, environment, diet, and diseases', Microorganisms, 7: 14.

Round 2

Reviewer 1 Report

Comments and Suggestions for Authors

Thanks for the modifications and explanations, I have no additional comments.

Reviewer 3 Report

Comments and Suggestions for Authors

The authors took into account the indications of the reviewers and/or explained the mistakes made in writing the manuscript. I believe that the work can be published.